# Multicenter Real-World Study: 432 Patients with Apalutamide in Metastatic Hormone-Sensitive Prostate Cancer

**DOI:** 10.3390/curroncol32030119

**Published:** 2025-02-21

**Authors:** Juan A. Encarnación Navarro, Virginia Morillo Macías, María Borrás Calbo, Isabel De la Fuente Muñoz, Antonio Lozano Martínez, Vicente García Martínez, Luis Fernández Fornos, Miriam Guijarro Roche, Osamah Amr Rey, Raquel García Gómez

**Affiliations:** 1Department Radiation Oncology, Hospital Clínico Universitario Virgen de la Arrixaca, 30120 Murcia, Spain; isabeldelafuente123@gmail.com (I.D.l.F.M.); antoniojlozano@live.com (A.L.M.); 2Faculty of Medicine, University of Murcia, 30100 Murcia, Spain; 3Murcian Institute of Biosanitary Research, 30120 Murcia, Spain; 4Department Radiation Oncology, Consorcio Hospitalario Provincial, 12002 Castellón, Spain; vmorill@gmail.com; 5Department Radiation Oncology, Hospital Universitario y Politécnico La Fe, 46026 Valencia, Spain; meryborras14@gmail.com; 6Department Radiation Oncology, Hospital General Universitario Santa Lucía, 30202 Cartagena, Spain; vgm20@hotmail.com; 7Department Radiation Oncology, Hospital Universitario San Juan, 03550 Alicante, Spain; lferfor@gmail.com (L.F.F.); guiroche@hotmail.es (M.G.R.); 8Department Radiation Oncology, Consorcio Hospital General Universitario, 46014 Valencia, Spain; radioterapia_hgv@gva.es; 9Department Radiation Oncology, Hospital Clínico Universitario, 15706 Valencia, Spain; raquelitagarcia@gmail.com

**Keywords:** prostate cancer, apalutamide, PSA decline, ultralow PSA

## Abstract

**Background:** The management of metastatic hormone-sensitive prostate cancer (mHSPC) has evolved significantly in recent years due to the introduction of androgen receptor-targeted agents (ARTAs). When used alongside androgen deprivation therapy (ADT), these treatments have shown improved oncological results and enhanced survival rates for patients with this condition. **Objectives:** The objective of this study was to describe the decline in prostate-specific antigen (PSA), the oncological outcomes, and the toxicity profile of mHSPC patients treated with apalutamide. **Materials and Methods:** Clinical data obtained from seven national hospitals were utilized between March 2021 and July 2024. PSA responses were collected at 3, 6, 12, 18, and 24 months, along with adverse effects reported by patients, dose reductions, or drug discontinuations. The association between PSA decline and progression-free survival (PFS) was evaluated with respect to metastasis volume, location, and timing of diagnosis. **Results:** A total of 432 patients were included, of whom 40% were de novo cases, and the greater part were classified as M1b. After one year, a reduction of more than 90% in PSA levels was observed in 88.2% of cases, with undetectable levels (≤0.2 ng/mL) achieved in 81.7% of them. The drug was discontinued in 76 patients (15.6%), with adverse effects reported in 7.8% (grade 3) and 1.9% (grade 4). Regarding PSA levels <0.02 ng/mL, promising results were observed, with ultralow PSA (UL2) achieved in 43% of cases at 6 months. **Conclusions:** This study revealed strong oncological outcomes, with rapid and profound PSA declines and drug safety consistent with emerging evidence. The distinctive finding of this study underscores the importance of a rapid and profound response (UL PSA) as a predictor of better oncological outcomes.

## 1. Introduction

Prostate cancer (PC) is now the second most prevalent malignancy in men globally, with an estimated 1.6 million new cases and around 366,000 deaths each year [1,2]. In Europe, the incidence is 10.8% of new cases, with around 268,490 cases and 34,500 deaths per year [3]. Despite the significant impact of these figures, PC shows a high five-year survival rate, exceeding 98%. In Spain, the statistics are similar, with an incidence of 12.3% and a mortality rate of 5.1% [4]. Of this total, approximately 4% are metastatic cases, which have a significantly worse prognosis compared to non-metastatic forms of PC [5,6].

The therapeutic response in metastatic hormone-sensitive prostate cancer (mHSPC) varies, influenced by the diverse nature of this patient group and their classification based on metastatic burden (high volume [HV] or low volume [LV]) and the timing of disease onset (de novo or metachronous) [7,8]. This classification has relevant prognostic implications, showing higher overall survival rates and a longer time to develop castration resistance in patients with low-volume metachronous mHSPC [9,10].

Prostate cancer management has evolved considerably over the past decades, transitioning from a disease primarily treated with androgen deprivation therapy (ADT) alone to a more nuanced and multimodal approach. The landscape of mHSPC treatment has been reshaped by the introduction of novel androgen receptor signaling inhibitors, offering improved outcomes for patients across different risk profiles [11]. However, despite these advancements, challenges remain in optimizing treatment sequencing, identifying patients most likely to benefit from specific therapies, and understanding the long-term impact of early intensification strategies. Additionally, emerging evidence highlights the role of molecular and genetic biomarkers in predicting response to treatment, emphasizing the need for a more personalized approach to therapy selection [12].

Its treatment has significantly evolved since the introduction of androgen deprivation therapy (ADT) [11]. While this approach has proven effective, European guidelines discourage its use as monotherapy [13], given that a significant percentage of patients develop castration resistance [14]. This fact has led to the emergence of new therapeutic agents such as apalutamide [15,16], abiraterone [17,18,19], enzalutamide [20,21], and darolutamide [22]. Notably, the TITAN clinical trial demonstrated that apalutamide can prolong metastasis-free survival (MFS) and improve overall survival (OS) compared to placebo, thus offering a key therapeutic option for this population [23]. However, there is a notable underrepresentation of certain patient groups, emphasizing the importance of complementing evidence from randomized clinical trials with analyses of real-world clinical practice data [24,25,26,27].

Despite the growing body of evidence supporting the efficacy of intensified treatment strategies, discrepancies persist in clinical decision-making, often influenced by patient comorbidities, regional treatment accessibility, and clinician preferences. Moreover, real-world data suggest that treatment patterns may diverge from those observed in clinical trials, particularly in older patients or those with significant comorbidities. Therefore, evaluating the effectiveness and safety of these agents in a real-world setting is essential to bridge the gap between controlled trial conditions and routine clinical practice. Addressing these gaps will provide a clearer understanding of how to optimize mHSPC management and ensure that the benefits of novel therapies are extended to broader patient populations.

The objective of this study is to comprehensively evaluate the efficacy and safety of apalutamide in patients with metastatic hormone-sensitive prostate cancer (mHSPC) within a real-world clinical setting. By analyzing treatment outcomes beyond the controlled conditions of clinical trials, this study aims to provide valuable insights into how apalutamide performs in diverse patient populations with varying clinical characteristics, comorbidities, and treatment histories. Furthermore, this research seeks to identify potential predictive factors associated with therapeutic response and disease progression, contributing to a more personalized approach to patient management. Understanding real-world effectiveness is essential for optimizing treatment strategies, addressing potential gaps in guideline-based care, and refining clinical decision-making. Ultimately, by reinforcing the role of apalutamide as a key therapeutic option, this study aspires to enhance patient outcomes and inform future clinical practice in the evolving landscape of mHSPC management.

## 2. Materials and Methods

### 2.1. Study Methodology

After approval by the ethics committee, this study included a retrospective cohort of patients diagnosed with mHSPC between March 2021 and July 2024. The sample was obtained from seven national hospitals (Murcia and Valencia regions). mHSPC was defined according to the CHAARTED criteria [28], with a more precise categorization based on the volume and timing of metastases, according to the Francini and Gravis classification [9,10].

### 2.2. Data Collection

Data were extracted from patient records and tracked until either the completion of follow-up or the occurrence of death, whichever came first. The evaluation included two categories of data: baseline characteristics and outcomes after drug initiation.

For baseline characteristics, individual records included information such as age, PSA levels, history of prior local treatment for PC, and metastasis volume and location.

After drug initiation, evaluations included PSA reductions >90% at 1, 3, 6, 12, 18, and 24 months, as well as adverse effects, dose reductions, or drug discontinuations. PSA levels were classified according to predefined ranges: >0.2, 0.02–0.2, and <0.02 ng/mL. Survival outcomes, such as progression-free survival (PFS), were evaluated based on predefined PSA ranges. Time-to-event data were measured in months, calculated from the start of treatment to the occurrence of an event (death or radiographic progression) or the most recent follow-up.

### 2.3. Statistical Analysis

A comprehensive descriptive analysis was conducted on patients’ baseline characteristics, adverse effects, and PSA response. Categorical variables were reported as percentages, while continuous variables were summarized using medians and ranges. The relationship between PSA reduction, metastatic burden, location, and timing of diagnosis was examined.

PSA decline was evaluated based on a 90% reduction, as well as achieving PSA levels within the ranges of 0.02–0.2 ng/mL (UL1 PSA) and ≤0.02 ng/mL (UL2 PSA) during the study period. The impact of PSA reduction, metastasis volume, location, and diagnosis timing on survival outcomes was assessed using the Kaplan–Meier method and Cox regression analysis.

All statistical tests were two-tailed, with significance defined as *p* < 0.05. Data analyses were performed using SPSS for Windows, version 25.0.

## 3. Results

### 3.1. Patient Characteristics

The retrospective study sample included 432 patients. Regarding the timing of metastasis, 40% presented at disease onset, while 60% were metachronous. The majority of patients were classified as low-volume (82.4%), and 32.17% received metastasis-directed therapy (MTD), primarily using SBRT on nodal lesions. A complete breakdown of baseline characteristics is provided in Table 1.

### 3.2. PSA Levels

PSA values were collected during follow-up at 3-month intervals, with an initial determination at 1 month. A 90% reduction in PSA levels at one year was observed in 89.6% of patients.

Comparisons of PSA data were made between groups categorized by disease volume and timing of onset (Table 2 and Table 3).

### 3.3. PSA Ultralow

In regard to undetectable PSA, we analyzed the percentage of patients achieving a PSA UL1 decline (>0.02–≤0.2 ng/mL), which was 43.7% at 3 months and 36.8% at 6 months. Significant differences in progression rates were observed at 3 months (0% vs. 4%, *p* = 0.027) and at 6 months (0.2% vs. 3.7%, *p* = 0.030) between patients who reached UL2 levels (≤0.02 ng/mL).

### 3.4. Progression-Free Survival (PFS)

In the survival analysis, we observed that both a greater than 90% reduction in PSA levels and achieving PSA levels < 0.2 ng/mL were associated with a reduced risk of progression. This association was statistically significant at all time points evaluated. At 12 months, the PFS for the group with PSA < 0.2 ng/mL was 95.1%, compared to 62% for PSA > 0.2 ng/mL, with a median PFS of 41.3 months versus 31.9 months, respectively (*p* = 0.000) (Figure 1). In multivariate analysis, only UL2 levels at 12 months were found to be an independent protective predictor against progression (HR −2.999, CI 0.006–0.450, *p* = 0.008).

Similarly, at 18 months, a significant difference in PFS was observed when PSA levels were UL2 (*p* = 0.048).

In reference to disease volume, a significant difference in PFS was also determined between the two groups, with a median of 40.1 months in the low-volume group compared to 33.9 months in the high-volume group (*p* = 0.018) (Figure 2).

Outcomes were analyzed in patients undergoing next-generation imaging (NGI) or conventional imaging studies, with a significant increase in PFS observed in favor of the NGI group (89.9% vs. 84.8%, *p* = 0.045).

### 3.5. Toxicity

In the study sample, an overall prevalence of adverse events of 44.9% was identified, with severe grades occurring in 9.7% (7.8% G3 and 1.9% G4). The most frequent events included maculopapular rash, asthenia, and hypertension, leading to treatment discontinuation in 15.6% of cases. In multivariate analysis, treatment discontinuation increased the risk of disease progression (HR 11.102, CI 4.157–29.646, *p* = 0.000). Regarding the observed side effects, skin rash was one of the most common, affecting 15.9% of patients in this study. Within this group, a smaller percentage of more severe cases, specifically grade 3–4, was recorded, accounting for 5.3% of patients.

## 4. Discussion

Traditionally, ADT has been the standard therapy for mHSPC patients until 2015, when its efficacy variability and primary resistance were identified [29]. The STAMPEDE trial demonstrated prolonged responses to ADT [30], but identifying these patients remains an unresolved issue. This is why European guidelines discourage its use as monotherapy for this patient profile [8,11,13,29].

Apalutamide is a non-steroidal oral antiandrogen that binds directly to the ligand-binding domain of the androgen receptor (AR), inhibiting its translocation, DNA binding, and AR-driven transcription [31]. Studies have shown that combining androgen deprivation therapy (ADT) with apalutamide enhances AR inhibition, leading to a more effective blockade of androgen signaling and improved oncological outcomes in patients with metastatic hormone-sensitive prostate cancer (mHSPC) and high-risk non-metastatic castration-resistant prostate cancer (nmCRPC) [15,32]. In both pivotal trials, conventional imaging techniques such as CT scans and bone scans were used for diagnosis. However, next-generation imaging (NGI) revealed metastases in 58% of patients who were initially classified as high-risk non-metastatic based on conventional imaging [33]. For this reason, NGI more accurately categorizes patients, positively impacting disease control.

In our analysis, patients diagnosed through NGI showed greater PFS compared to those diagnosed using conventional methods. This finding is significant, as NGI allows for earlier and more individualized disease detection [34,35].

In 2019, the phase III TITAN clinical trial reported increased radiographic progression-free survival (rPFS) and overall survival (OS) with an acceptable safety profile in mHSPC patients treated with apalutamide. The included population was diagnosed with conventional imaging and excluded exclusive nodal involvement (M1a) and significant cardiovascular comorbidities [15]. In 2021, the final analysis confirmed the oncological benefit of apalutamide treatment [15,32]. Additionally, a post hoc analysis of the SPARTAN trial revealed that PSA response was linked to both short- and long-term oncological benefits, reinforcing the value of PSA monitoring in clinical practice. Likewise, a separate post hoc analysis of the TITAN trial identified a strong correlation between PSA reduction and improved oncological outcomes, further validating PSA decline as a predictive marker for survival [36].

However, the evidence from clinical trials, despite their methodological rigor, cannot fully reflect the clinical reality of this disease. Trials tend to include homogeneous patient populations, often with optimal health profiles, which may not represent everyday practice, where the population includes older patients with comorbidities [37].

Survival analysis in our cohort revealed a significant association between PSA levels and patients’ clinical progression. In particular, patients maintaining PSA levels below 0.2 ng/mL during follow-up demonstrated higher PFS rates. This relationship was significant at all analyzed time points (*p* < 0.05), suggesting that sustained PSA reduction to undetectable levels could serve as a positive prognostic marker in oncological follow-up. In our series, achieving UL2 PSA levels at 12 months was an independent protective factor against progression.

Additionally, in the group of patients achieving a greater than 90% reduction in initial PSA levels, an increase in PFS was also observed. Our series showed a response in PSA kinetics, with a 90% PSA reduction and ultralow levels (<0.2 ng/mL) in a significant percentage of patients.

A post hoc exploratory analysis of the TITAN study revealed that patients treated with apalutamide achieved higher UL1 and UL2 PSA levels than the placebo group. This was associated with significantly longer rPFS. This recent study demonstrates the feasibility of detecting ultralow PSA levels, which could be considered an early indicator of AR inhibition strength. This was particularly evident in prostatectomized patients, where ultralow PSA levels were associated with a lower risk of biochemical progression [38].

Since the relationship between apalutamide-mediated AR inhibition and clinical outcomes is well established, monitoring ultralow PSA levels could identify mHSPC patients who would benefit most from this drug.

Therefore, we infer that an early and deeper decline in PSA levels is associated with better oncological outcomes. These results highlight the clinical value of PSA kinetics as a marker of therapeutic efficacy.

Another differentiating aspect of our series is the inclusion of patients with exclusive M1a metastases, a group not considered in the TITAN trial [15,16]. In this study, it was observed that 35.1% of patients (152) presented M1a stages, a subtype of metastasis that was not included in the TITAN trial. The inclusion of patients with M1a represents a significant difference compared to the design of other key studies in this field, allowing for a broader evaluation of therapeutic options in a more diverse population. In the TITAN trial, patients with M1a metastases were not included as a specific subgroup, limiting the extrapolation of its results to this population, which might benefit from a differentiated therapeutic approach due to the unique biology of these metastases. Preliminary data from our series suggest that apalutamide combined with ADT could provide significant clinical benefits for patients with M1a metastases. This highlights the importance of conducting specific analyses for this subgroup, as nodal involvement may exhibit a different response to this type of hormonal therapy. Including patients with exclusive nodal metastases in our series not only underscores the differences with the TITAN trial but also provides valuable information about apalutamide’s efficacy in this subgroup.

In terms of observed side effects, skin rash emerged as one of the most frequent, affecting 15.9% of patients in this study. Among this group, a smaller proportion experienced more severe forms, specifically grade 3–4, which accounted for 5.3% of patients, something similar to the TITAN study [15,16]. To address this adverse reaction, a variety of therapeutic strategies were employed, including the use of topical corticosteroids and, in some cases, antihistamines to manage symptoms. Most patients with skin rashes were treated symptomatically, with early interventions aimed at preventing progression to more severe stages. Additionally, symptom progression was closely monitored, with treatment doses adjusted or temporarily suspended in more severe cases, ensuring adequate recovery without significantly hindering the overall treatment plan.

These findings could help optimize therapeutic strategies in clinical practice. In our study, a higher percentage of patients achieved UL1 and UL2 PSA levels, with results superior to those reported in the TITAN trial. Furthermore, the drug demonstrated an acceptable safety profile, with adverse event rates comparable to those observed in pivotal studies [15,16,32].

Moreover, the real-world nature of this study is another significant strength, as it provides evidence gathered practically and directly from the patient population, reflecting the conditions and variability inherent in daily clinical practice. Unlike randomized controlled trials (RCTs), which may be more rigid and controlled, real-world studies allow for a broader and more representative evaluation of how treatments and diagnostic methods perform in a diverse population. This provides greater applicability of the findings to general clinical practice, offering more useful information for physicians in their decision-making and providing a more realistic framework for the effectiveness of ultralow PSA in monitoring prostate cancer recurrence.

### Limitations

One of the main limitations of this study lies in the collection of patient data, which, although successfully gathered for all cases, presents the inherent restriction of having been obtained retrospectively. This approach inevitably carries a risk of bias, as the quality and accuracy of the information depend on pre-existing records that may not have been specifically designed to address the research questions posed in this study. Furthermore, the fact that the data were not collected prospectively prevents a more rigorous control over the studied variables, which could influence the interpretation of the results and the external validity of this study. This retrospective bias may lead to the unintentional omission of relevant information or reliance on records that, in some cases, may be incomplete or subject to systematic errors. Despite efforts to minimize these potential distortions—such as cross-validation of the information and comparison with different available sources—the retrospective nature of the data remains a limitation to consider when extrapolating the findings to other populations or clinical contexts.

## 5. Conclusions

Evaluating the efficacy and safety of apalutamide in our population confirms the results of previous clinical trials, providing relevant information on the characteristics and treatment of these patients in a real-world clinical setting. Thus, apalutamide is established as a key drug in mHSPC management.

Our results indicate that apalutamide induces a rapid and significant PSA response (UL PSA), predicting better oncological outcomes. Moreover, the data suggest that this drug could be an effective therapeutic option for the M1a population, though more extensive studies are needed to confirm this.

## Figures and Tables

**Figure 1 curroncol-32-00119-f001:**
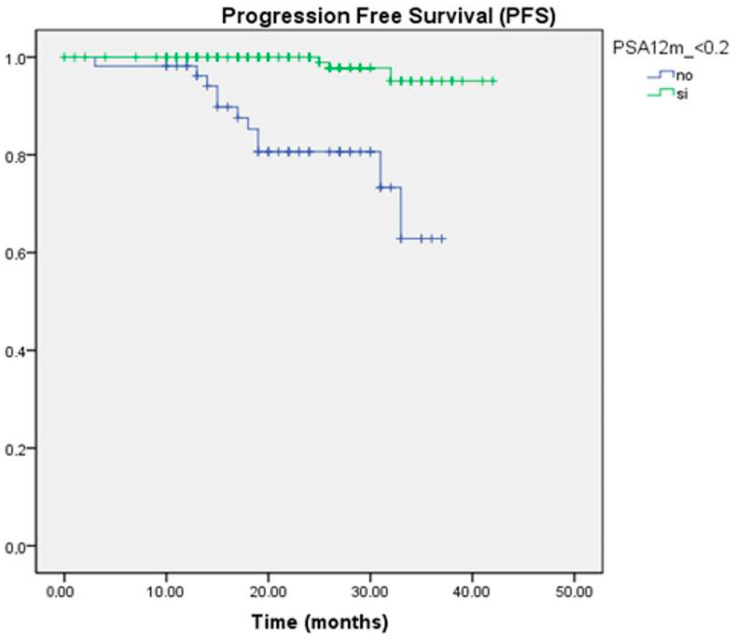
Kaplan–Meier survival curve for PFS according to PSA ranges at 12 months.

**Figure 2 curroncol-32-00119-f002:**
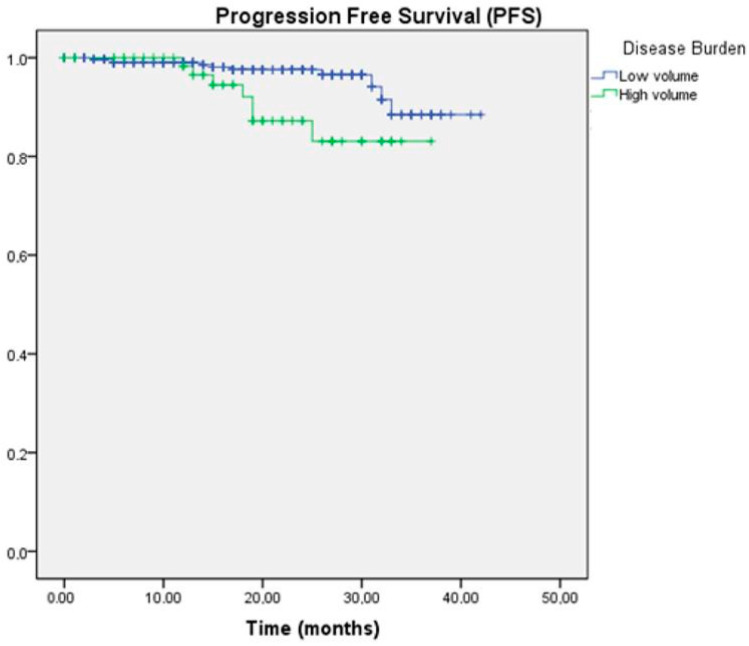
Kaplan–Meier survival curve for PFS according to disease volume.

**Table 1 curroncol-32-00119-t001:** Baseline characteristics of the patients.

	n: 432
Median age at the start of treatment	72 years (40–87)
Diagnostic PSA (median)	8.39 ng/mL (0.03–1410)
Baseline PSA prior to apalutamide (median)	23.88 ng/mL (0.03–2587)
mHSPC (proportion, n)	
M1a	35.1% (152)
M1b	54.9% (237)
M1c	10% (43)
SBRT (proportion, n)	
Bone metastasis	58.28% (81)
Lymph node metastasis	30.93% (43)
Others	10.79% (15)

**Table 2 curroncol-32-00119-t002:** PSA response rate according to disease volume.

	Low Volume		High Volume	
	PSA Response >90% (% of Patients)	Undetectable PSA (% of Patients)	PSA Response >90% (% of Patients)	Undetectable PSA (% of Patients)
1° month	78.6	51.21	51.5	33.3
3° month	93.5	86.25	85.7	46
6° month	94.9	87.1	86.2	51
12° month	92.3	87.5	83.6	59.2
18° month	90.4	87	82.1	64.3
24° month	89	85.6	75.8	58.6

**Table 3 curroncol-32-00119-t003:** PSA response rate according to the timing of metastasis onset.

	M1 De Novo		M1 Metachronous	
	PSA Response >90% (% of Patients)	Undetectable PSA (% of Patients)	PSA Response >90% (% of Patients)	Undetectable PSA (% of Patients)
1° month	63.9	50	77.1	55.9
3° month	88	60.7	89.3	80.9
6° month	89.9	71.7	92.1	86.5
12° month	88.8	75.9	90.5	84.7
18° month	93	81.9	88.3	79.6
24° month	90.3	79.16	86.4	77

## Data Availability

The data supporting the reported results of this study are available upon request from the corresponding author. Due to privacy or ethical restrictions, the data are not publicly available.

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
