# Peer review of "Multicenter Real-World Study: 432 Patients with Apalutamide in Metastatic Hormone-Sensitive Prostate Cancer"

_curroncol, 2025, doi:10.3390/curroncol32030119_

Round 1
Reviewer 1 Report
Comments and Suggestions for Authors
Thank you for preparing and submitting a good paper.
I think it is a vivid report of patients encountered in real practice.
There are several things that I would like the authors to add to the paper.
- The authors mentioned the importance of NGI. Please provide a more detailed table showing how many patients were tested for NGI and what the results were.
- Another topic of interest in real practice is side effects. Please provide more details on side effects, especially skin side effects, and indicate how they were treated and when patients were dropped.
- The authors wrote that they had experience with patients with M1a disease, which was not included in other studies. Please list and analyze the data in more detail in the text and insert it.
I appreciate the authors' hard work and effort.
Author Response
Thank you very much for your thorough review and valuable feedback. We have carefully addressed all the issues raised and made the necessary revisions. We appreciate your insights, which have contributed to improving the quality of the manuscript.
Thank you for preparing and submitting a good paper.
I think it is a vivid report of patients encountered in real practice.
There are several things that I would like the authors to add to the paper.
- The authors mentioned the importance of NGI. Please provide a more detailed table showing how many patients were tested for NGI and what the results were.
- Another topic of interest in real practice is side effects. Please provide more details on side effects, especially skin side effects, and indicate how they were treated and when patients were dropped.
- The authors wrote that they had experience with patients with M1a disease, which was not included in other studies. Please list and analyze the data in more detail in the text and insert it.
Replies to comments
- We appreciate your comments and the opportunity to improve our manuscript. In response to your request regarding next-generation testing (NGI), we can confirm that a total of 255 patients were evaluated using this technique. However, we do not have individual data on the specific results for each patient. Our comparison focused on the proportion of patients who underwent next-generation testing compared to those who were evaluated using conventional tests.
- Regarding point 2, we have added more details on side effects, especially skin-related ones, including their management and the timing of patient withdrawal from the study.
- Regarding point 3, we have expanded the information on our experience with patients with M1a disease, providing a more detailed analysis in the text and incorporating the corresponding data.
We appreciate your observations, which have been very helpful in improving the clarity and quality of our work.
Sincerely,
Juan Antonio Encarnación.
Reviewer 2 Report
Comments and Suggestions for Authors
The authors presented the retrospective study on PSA changes, the oncological outcomes, and the toxicity profile of mHSPC patients treated with apalutamide.
Please explain several doubts that were listed below:
Lines 74-5 page 2: please include the specific definition that you cited, especially as for the volume and timing.
Table 1 Please translate: OSEA, GANGLIONAR, OTRAS.
Table 1 Please double check diagnostic PSA and baseline PSA values – is it truly median? Are the ranges in the commas? If so, please add this information.
Table 1 Can you characterize the population studied further? e.g. any comorbidities? Pathological features, e.g. Gleason? Any previous treatments (as you stated in page line 82)? You mentioned also that 60% of M+ cases were metachronous. Can you specify the OS or the timelines of the treatment?
Table 3 Please translate: METACRONICO.
Page 4, lines 128-9; page 5 lines 138-9: Please round the numbers, e.g. 41.3 or 31.9, etc.
Page 7 line 224 Did you have access to all necessary data? As for the sample (n=432), did you collect all the data in all cases? The flowchart or short comment would be advisable in either methodology or limitations section.
Page 7 lines 226-230: If your retrospective study results confirmed the previous findings coming from RCT, what is the major conclusion and novelty of your paper?
Please change the name of supplementary files as these are the major (basic) figures and tables used in the text.
Author Response
Thank you for your valuable comments. We have addressed all the points raised. Additionally, we have included a note in the limitations section regarding the collection of patient data. Although we were able to gather data for all patients, we acknowledge that this retrospective data collection presents a limitation, as it may introduce bias. This retrospective nature of the data could affect the accuracy and interpretation of the results, and we have made sure to highlight this in the revised manuscript.
The authors presented the retrospective study on PSA changes, the oncological outcomes, and the toxicity profile of mHSPC patients treated with apalutamide.
Please explain several doubts that were listed below:
- Lines 74-5 page 2: please include the specific definition that you cited, especially as for the volume and timing.
- Table 1 Please translate: OSEA, GANGLIONAR, OTRAS.
- Table 1 Please double check diagnostic PSA and baseline PSA values – is it truly median? Are the ranges in the commas? If so, please add this information.
- Table 1 Can you characterize the population studied further? e.g. any comorbidities? Pathological features, e.g. Gleason? Any previous treatments (as you stated in page line 82)? You mentioned also that 60% of M+ cases were metachronous. Can you specify the OS or the timelines of the treatment?
- Table 3 Please translate: METACRONICO.
- Page 4, lines 128-9; page 5 lines 138-9: Please round the numbers, e.g. 41.3 or 31.9, etc.
- Page 7 line 224 Did you have access to all necessary data? As for the sample (n=432), did you collect all the data in all cases? The flowchart or short comment would be advisable in either methodology or limitations section.
- Page 7 lines 226-230: If your retrospective study results confirmed the previous findings coming from RCT, what is the major conclusion and novelty of your paper?
- Please change the name of supplementary files as these are the major (basic) figures and tables used in the text.
Replies to comments
We appreciate your observations, which have been very helpful in improving the clarity and quality of our work.
1-2-3-5-6-9: We would like to inform you that points 1, 2, 3, 5, 6, and 9 have been addressed and resolved in the text. Your clarifications have been very helpful in improving the accuracy and clarity of our work.
4- In response to your observation regarding the characterization of the studied population (Table 1), we did not collect information on comorbidities or Gleason score for this article, so we were unable to address this point.
7- In response to your observation regarding data availability (Page 7, line 224), we have included a limitations section where we explain this point in detail. In this section, we clarify data access and the collection of information for the sample of 432 patients.
8- In response to your observation regarding the novelty and main conclusion of our study (Page 7, lines 226-230), we have added an explanation in the text highlighting the unique value of our work. In particular, we emphasize that the real-world nature of this study is a significant strength, as it provides evidence obtained directly from the patient population, reflecting the conditions and variability of daily clinical practice.
Unlike randomized controlled trials (RCTs), which tend to be more rigid and controlled, our study allows for a broader and more representative evaluation of how treatments and diagnostic methods perform in a diverse population. This enhances the applicability of the findings to general clinical practice, providing valuable information for medical decision-making and a more realistic framework for assessing the effectiveness of ultra-low PSA in monitoring prostate cancer recurrence.
We appreciate your observations, which have been essential in improving the clarity and relevance of our work.
Sincerely,
Juan Antonio Encarnación.
Round 2
Reviewer 2 Report
Comments and Suggestions for Authors
The authors responded to all my comments.